# Long-Term Rotary Tillage and Straw Mulching Enhance Dry Matter Production, Yield, and Water Use Efficiency of Wheat in a Rain-Fed Wheat-Soybean Double Cropping System

**DOI:** 10.3390/plants14152438

**Published:** 2025-08-06

**Authors:** Shiyan Dong, Ming Huang, Junhao Zhang, Qihui Zhou, Chuan Hu, Aohan Liu, Hezheng Wang, Guozhan Fu, Jinzhi Wu, Youjun Li

**Affiliations:** College of Agriculture, Henan University of Science and Technology, Luoyang 471023, China; dongshiyan@stu.haust.edu.cn (S.D.); huangming@haust.edu.cn (M.H.); zhangjunhao@stu.haust.edu.cn (J.Z.); zhouqihui@stu.haust.edu.cn (Q.Z.); hc23@stu.haust.edu.cn (C.H.); 230320170987@stu.haust.edu.cn (A.L.); wanghezh@haust.edu.cn (H.W.); gzfu@haust.edu.cn (G.F.)

**Keywords:** straw mulching, wheat-soybean double-cropping, water productivity, grain yield

## Abstract

Water deficiency and low water use efficiency severely constrain wheat yield in dryland regions. This study aimed to identify suitable tillage methods and straw management to improve dry matter production, grain yield, and water use efficiency of wheat in the dryland winter wheat–summer bean (hereafter referred to as wheat-soybean) double-cropping system. A long-term located field experiment (onset in October 2009) with two tillage methods—plowing (PT) and rotary tillage (RT)—and two straw management—no straw mulching (NS) and straw mulching (SM)—was conducted at a typical dryland in China. The wheat yield and yield component, dry matter accumulation and translocation characteristics, and water use efficiency were investigated from 2014 to 2018. Straw management significantly affected wheat yield and yield components, while tillage methods had no significant effect. Furthermore, the interaction of tillage methods and straw management significantly affected yield and yield components except for the spike number. RTSM significantly increased the spike number, grains per spike, 1000-grain weight, harvest index, and grain yield by 12.5%, 8.4%, 6.0%, 3.4%, and 13.4%, respectively, compared to PTNS. Likewise, RTSM significantly increased the aforementioned indicators by 14.8%, 10.1%, 7.5%, 3.6%, and 20.5%, compared to RTNS. Mechanistic analysis revealed that, compared to NS, SM not only significantly enhanced pre-anthesis and post-anthesis dry matter accumulation, and pre-anthesis dry matter tanslocation to grain, but also significantly improved pre-sowing water storage, water consumption during wheat growth, water use efficiency, and water-saving for produced per kg grain yield, with the greatest improvements obtained under RT than PT. Technique for Order Preference by Similarity to Ideal Solution (TOPSIS) analysis confirmed RTSM’s yield superiority was mainly ascribed to straw-induced improvements in dry matter and water productivity. In a word, rotary tillage with straw mulching could be recommended as a suitable practice for high-yield wheat production in a dryland wheat-soybean double-cropping system.

## 1. Introduction

Wheat (*Triticum aestivum* L.)—a globally essential staple crop and the third-largest food crop in China [1], contributes approximately 20% of the caloric intake and protein supply for mankind [2]. In China, approximately one-third of wheat is planted in dryland areas [3,4]. In Henan, wheat constitutes over one-quarter of annual production in China, and dryland wheat also comprises one-third of the region’s wheat acreage [1]. In dryland wheat production regions, natural precipitation serves as the sole water source for wheat growth. Thus, the limited rainfall and the mismatch between the rainy summer season and the wheat-growing season severely constrain the yield formation of wheat. Dry matter accumulation underpins the material basis of crop yield formation, with its accumulation, translocation, and distribution closely linked to crop yield and water use efficiency (WUE) [5]. Therefore, exploring agronomic techniques to improve the dry matter characteristics in plants for enhancing the yield and water use efficiency is of significance in promoting dryland wheat yield and ensuring food security in China and worldwide.

Both plowing and rotary tillage have been widely applied to improve crop dry matter production and enhance crop yield and WUE [5,6,7]. However, the impacts of these two tillage methods on soil physical and chemical properties, dry matter production, grain yield, and WUE were different. In general, plowing alleviates subsoil compaction, reducing bulk density and improving porosity, but may accelerate soil organic matter mineralization [8]. Rotary tillage enhances the porosity and water retention in topsoil (0–20 cm), while preserving subsoil structure [9]. Zheng et al. [10] found that plowing increased wheat dry matter production and WUE compared to rotary tillage. Shi et al. [6] employed the Agricultural Production Systems simulator (APSIM) model to demonstrate that plowing significantly enhanced simulated average annual soil water storage (by 8.4%), above-ground dry matter accumulation (18.4%), grain yield (25.5%), and WUE (22.3%) compared to rotary tillage. However, some studies indicated that rotary tillage exhibits superior soil water retention and moisture conservation capabilities, which contribute to improved soil water supply during the later wheat growth period and result in a higher wheat yield compared to plowing [11]. Kan et al. [7] also found that rotary tillage increased the spatial and temporal root distribution as well as photosynthetic activity at the flowering stage, resulting in an increase of 12.0% in grain yield compared to plowing. These studies illustrated that the effect of rotary tillage and plowing on wheat productivity is inconsistent and requires further investigation.

Crop straw contains a large amount of organic matter and nutrients such as N, P, and K [12,13]. Straw mulching has been proven to optimize soil conditions (water, nutrients, aeration, and temperature) through comprehensive effects including mitigating soil erosion, buffering temperature fluctuations, and reducing water evaporation, resulting in enhancing the growth and physiological characteristics of root and above-ground parts, thereby affecting WUE, dry matter production, and ultimately enhancing grain yield [14,15,16,17,18,19,20]. For instance, Huang et al. [19] reported that straw mulching improved grain yield, WUE for grain yield (WUE_r_), and WUE for aboveground biomass (WUE_b_) by 6.9%, 11.3%, and 16.5%, respectively, compared to no straw mulching. Similarly, Zhang et al. [20] found that, in the Loess Plateau, straw mulching increased wheat yield by 13.3–23.0%, and WUE by 15.2–18.0% over the three years due to increasing the soil water content by 0.7–22.5% and reducing the 2–10 days for soil moisture less than 60% field capacity compared to no straw mulching [21]. Ram et al. [22] also reported straw mulching treatments (2, 4, and 6 t ha^−1^) increased WUE by 14.7–34.2% compared to no straw mulching across different irrigation levels. However, the effectiveness of straw mulching was region specific and can be influenced by initial soil moisture conditions. Many studies showed that the interaction between tillage methods and straw management has an effect on the crop dry matter characteristics, yield, and WUE [23,24,25]. Some studies showed that plowing combined with straw mulching improved soil water storage and root growth, thereby promoting the characteristics of dry matter translocation and distribution and significantly increasing crop yield [23,24]. Liu et al. [25] demonstrated that rotary tillage with straw returning significantly increased dry matter accumulation by 28.1%, grain yield by 17.8%, and WUE by 27.9%, compared to plowing without straw returning. In addition, compared to the mono-cropping system, the double-cropping system with intensive cultivation accelerates straw decomposition due to frequent tillage and shorter fallow periods, enhancing short-term nutrient release [26]. However, the double-cropping system also exacerbates inter-crop water competition by reducing soil moisture retention capacity, particularly in arid regions [27]. Above all, the previous studies were mainly focused on the mono-cropping system in drylands, and the research indicators mainly concentrated on soil moisture and thermal characteristics, physical and chemical properties, crop yield and resource efficiency, and physiological and biochemical traits. However, the effects of tillage methods and straw management interaction on dry matter production, and the effect of straw mulching on water-saving and yield improvement in rain-fed wheat were still not well understood.

The analytical system initially employs Partial Least Squares Path Modeling (PLSPM) for pathway analysis, with model reliability verifiable through 500 bootstrap iterations [28,29]. Model fit is assessed using the standardized root mean square residual and comparative fit index, while multicollinearity is controlled via variance inflation factors. This analytical approach enables rigorous quantification of causal relationships and interaction mechanisms among variables, thereby providing a scientific basis for crop production system research [30]. Building upon this foundation, the TOPSIS methodology incorporates entropy weighting—where weights are automatically assigned based on information entropy, with indicators exhibiting greater data variability receiving higher weights and vector normalization being applied to eliminate dimensional effects without altering the fundamental ranking relationships. This dual approach ensures objective evaluation through calculation of relative distances between treatment alternatives and the ideal solution [28,29,30,31].

The winter wheat–summer soybean (hereafter referred to as wheat-soybean) double-cropping system is an important planting mode in China and worldwide. This system enhances soil properties and the micro-environment for crop growth by increasing organic matter content, promoting biological nitrogen fixation, preventing soil erosion, and reducing downward movement of soil moisture, while also suppressing weeds and diseases and improving nutrient cycling through soybean cultivation, thereby affecting wheat productivity [32,33]. However, the impacts of tillage methods and straw management on soil moisture, dry matter accumulation, translocation, distribution, as well as yield and WUE of wheat in the dryland wheat-soybean double-cropping system remained scarce. Therefore, this study aims to study these gaps based on a long-term field experiment with two tillage methods—plowing (PT) and rotary tillage (RT)—and two straw management methods—no straw mulching (NS) and straw mulching (SM). The objectives were to (1) assess the effects of tillage methods and straw management on wheat yield and its mechanism in terms of dry matter characteristics, soil moisture, and WUE; (2) comprehensively evaluate the factor contribution of wheat yield under tillage methods and straw management using Partial Least Squares Path Modeling (PLSPM) and Technique for Order Preference by Similarity to Ideal Solution (TOPSIS) methods; (3) provide theoretical and technical insights for improving dry matter production, grain yield, and the WUE of wheat in a dryland wheat-soybean double-cropping system.

## 2. Results

### 2.1. Yield and Yield Components

The experimental years, straw management, and the interaction of tillage methods and straw management (except for spike number) significantly affected the grain yield, yield components, and harvest index (Table 1). Compared to RTNS, PTNS increased 1000-grain weight and grain yield by 1.4% and 6.2% (375.5 kg ha^−1^), respectively, over the four years. Compared to NS, SM significantly improved wheat yield and yield components under the same tillage method. Specifically, the grain yield, spike number, grain number per spike, 1000-grain weight, and harvest index increased by 10.5% (669.1 kg ha^−1^), 10.4%, 6.4%, 4.4%, and 1.8%, respectively, under PT; and by 20.5% (1229.6 kg ha^−1^), 14.8%, 10.1%, 7.5%, and 3.6%, respectively, under RT over the four years. Considering the interaction effects, the grain yield in all four years followed the order RTSM > PTSM > PTNS > RTNS. There were no significant differences between RTSM and PTSM except for 1000-grain weight. However, both treatments were significantly higher than PTNS and RTNS.

### 2.2. Dry Matter Accumulation, Translocation, and Distribution

#### 2.2.1. Dry Matter Accumulation at Different Growth Stages

Figure 1 indicates that the experimental years, tillage methods (except for jointing stage), and straw management significantly influenced the dry matter accumulation of wheat. Over the four years, RTNS increased dry matter accumulation by 5.4% at the anthesis stage, but reduced it by 6.2% at the maturity stage, compared to PTNS. RTSM resulted in an average increase of 7.2% at the anthesis stage compared to PTSM. Under the same tillage method, straw mulching significantly increased dry matter accumulation across all growth stages and years. Over the four years, PTSM increased dry matter accumulation by 8.3% (1003.8 kg ha^−1^) at the anthesis stage and 8.6% (1860.9 kg ha^−1^) at the maturity stage compared to PTNS. RTSM increased dry matter accumulation by 9.7%, 10.2%, and 16.9%, with the values of 435.0, 1448.4, and 3146.6 kg ha^−1^ at the jointing, anthesis, and maturity stages, respectively, compared to RTNS. However, there was no significant difference between PTSM and RTSM.

#### 2.2.2. Characteristics of Translocation of Pre-Anthesis Dry Matter and Accumulation of Post-Anthesis Dry Matter

As shown in Table 2, the experimental years, tillage methods (excluding pre-anthesis translocation), and straw management significantly affected the dry matter characteristics of pre-anthesis translocation and post-anthesis accumulation. Over the four years, compared to NS, SM increased the pre-anthesis dry matter translocation and post-anthesis dry matter accumulation by 12.0% and 9.5%, respectively, under plowing, and by 5.2% and 49.8% under RT. However, compared to NS, SM decreased the pre-anthesis dry matter translocation rate and its contribution to grain by 4.7% and 15.1%, respectively, under RT. Compared to RTNS, PTNS increased the post-anthesis dry matter accumulation and its contribution to grain by 64.1% and 54.1%, respectively. RTSM increased the translocation amount, translocation rate, and contribution to grain of pre-anthesis dry matter by 25.0%, 16.1%, and 21.5%, respectively, compared to PTSM. Conversely, PTSM decreased the post-anthesis dry matter accumulation and its contribution to grain by 19.9% and 23.0%, respectively, compared to RTSM. RTNS also significantly increased the translocation amount, translocation rate, and contribution to grain of pre-anthesis dry matter, compared to PTNS.

#### 2.2.3. Dry Matter Distribution at Maturity

The experimental years and straw management also significantly affected wheat dry matter distribution at maturity, while tillage methods only influenced the dry matter distribution in stem + leaves and the percentage in grains (Table 3). Under NS, compared to RT, PT significantly increased the dry matter distribution in stem + leaves, glumes, and grains by 7.9%, 4.0%, and 6.4%, respectively. Under SM, compared to PT, RT increased grain dry matter distribution by 2.7%. The effects of straw management on dry matter distribution varied with tillage methods. Compared to NS, SM increased grain dry matter distribution proportion by −1.2–4.5%, with the significant differences were observed in 2 years under PT, it also increased grain dry matter distribution proportion by −0.2–6.3%, with the significant differences were observed in 3 years, while significantly decreased the percentage in stem + leaves in 3 years, with an average reduction of 3.2% over the four years under RT. RTSM obtained a similar dry matter distribution rate to PTSM. Both of thesetreatments significantly increased dry matter distribution and distribution percentage in grains, compared to PTNS and RTNS.

### 2.3. Soil Water Storage and WUE

#### 2.3.1. Soil Water Storage

Tillage methods and straw management can influence soil water storage in the 0–200 cm soil profile at sowing and maturity of wheat, with the notable differences across soil layers (Figure 2). Compared to RTNS, PTNS increased soil water storage at sowing by 7.3% (3.1 mm) in the 0–100 cm soil layer, whereas it decreased by 5.6% (2.3 mm) in the 100–200 cm soil layer. At maturity, the effects of different treatments on soil water storage varied with soil depth. In the 0–40 cm soil layer, SM significantly increased soil water storage compared to NS, with an increase of 25.4% (7.6 mm) under PT and 29.5% (7.6 mm) under RT. However, in the 40–200 cm soil layer, NS had higher soil water storage relative to SM, with an increase of 4.4% (2.2 mm) under PT and 2.2% (0.9 mm) under RT. Despite these trends, no significant differences were observed between the two tillage methods in the middle and deeper soil layers under the same straw management. Overall, the effect of straw mulching on soil water storage was greater than that of tillage methods, particularly in the upper soil layers.

#### 2.3.2. WUE

Figure 3A indicates that tillage methods had no significant impact on water consumption during the growing season (ET). However, under the same tillage method, SM significantly increased ET compared to NS, with an increase of 23.1 mm and 32.4 mm under PT and RT, respectively, over the four years. Figure 3B shows that under NS, PT significantly improved WUE by 1.0 kg ha^−1^ mm^−1^ compared to RT over the 4 years, with significant increases observed in 2 years. Likewise, under the same tillage method, SM significantly increased WUE by 0.7 kg ha^−1^ mm^−1^ under PT and 1.9 kg ha^−1^ mm^−1^ under RT compared to NS.

### 2.4. Effect of Straw Mulching on Water-Saving and Yield Improvement Under Different Tillage Methods

Further analysis revealed that the effects of straw mulching on water-saving and yield improvement of wheat varied depending on tillage methods (Table 4). Under different years and tillage methods, straw mulching consistently resulted in positive improvement in pre-sowing water storage, yield, and dry matter, and the positive values for the water-saving amount and rate per kg yield, and yield improvement amount and rate per mm water consumption. The effectiveness of straw mulching on these indicators under RT was more pronounced than PT. Specifically, the pre-sowing water storage, yield, and dry matter improvement under RT were significantly higher than those under PT in 2, 4, and 2 years, with a 4-year average increase of 13.9 mm, 554.9 kg ha^−1^, and 1261.1 kg ha^−1^, respectively. Similarly, the water-saving amount and rate per kg yield, as well as the yield-improvement amount and rate per mm of water consumption under RT were all higher than PT in 2 years (*p* < 0.05), with a 4-year average increase of 231.3% (3.7 mm), 196.8% (6.1%), 216.7% (1.3 kg ha^−1^), and 162.3% (21.1%), respectively.

### 2.5. Correlation and Path Model

Regression analysis (Figure 4) revealed a significant linear relationship between water consumption (ET) and wheat grain yield (except for 2017–2018). ET also exhibited a significantly positive correlation with dry matter accumulation over the four years. These results underscored the critical role of water consumption in improving wheat productivity. However, the water–yield relationships varied across different growing seasons. Conversely, the aggregated analysis demonstrated a significant positive correlation between water consumption and dry matter accumulation across years.

PLS-PM analysis further indicated that straw mulching significantly influenced wheat yield through multiple pathways (Figure 5A,B). Compared to NS, SM had a pronounced positive effect on yield (path coefficient, PC = 0.308, *p* < 0.01). Higher WUE (WUE) significantly contributed to improvements in both yield and dry matter accumulation, with path coefficients of PC = 0.236 (*p* < 0.01) and PC = 0.930 (*p* < 0.01), respectively. Additionally, the increases in dry matter accumulation were found to directly contribute to yield improvement (PC = 0.701, *p* < 0.01).

TOPSIS analysis (Table 5) showed that the comprehensive evaluation value (di) under RTSM, PTSM, PTNS, and RTNS ranged from 0.68 to 0.81, 0.57 to 0.69, 0.38 to 0.47, and 0.24 to 0.29, respectively. RTSM consistently showed the highest di, while RTNS recorded the lowest values across the four years, clearly indicating that RTSM consistently outperformed the other treatments.

## 3. Discussion

### 3.1. Effects of Tillage Methods and Straw Management on Wheat Yield

Optimizing wheat yield in rain-fed regions is crucial not only to meet the growing food demand but also to promote sustainable agricultural practices [34]. Research has shown that plowing is more effective than rotary tillage in breaking the plowing pan, promoting root penetration and development, ensuring water and nutrient supply during the later stages of growth, and increasing the spike numbers, grains per spike, and thousand-grain weight in wheat, thereby enhancing yield [5,35,36]. This study demonstrated that the effect of tillage methods on wheat yield varies with straw management. Under no straw mulching, plowing increased yield by 2.6% and thousand-grain weight by 1.5% compared to rotary tillage. Conversely, under straw mulching, compared to plowing, rotary tillage increased yield by 6.2% over the four years, with average increases in thousand-grain weight and harvest index both at 1.6%. The observed yield enhancement may be mainly ascribed to rotary tillage-induced acceleration of straw decomposition through stimulation of extracellular lignocellulose hydrolase activities and enrichment of copiotroph taxa, which facilitate synchronized nutrient mineralization (particularly ammonium-N and labile carbon) and promote root proliferation and photosynthetic assimilation efficiency [23,37].

The comparison between straw mulching and no straw mulching treatments underscores the critical role of straw in enhancing wheat yield. Under both plowing and rotary tillage, straw mulching consistently enhanced yield, with average increases of 10.5% under plowing and 20.5% under rotary tillage (Table 1). These yield improvements may be ascribed to straw mulching-induced improvement in organic matter decomposition, which enhanced soil moisture retention and nutrient availability, and directly benefited grain filling and 1000-grain weight [15,17,38]. The substantial yield gain under RTSM was mainly due to the synergistic effects of improved soil moisture and nutrient cycling [39]. This demonstrated that straw mulching is essential for maintaining soil moisture, particularly in rain-fed cropping systems where water stress is a prominent limiting factor for yield formation. Under no straw mulching, rapid soil moisture loss, and greater temperature fluctuations further stress the crop, while the reduced organic matter content limits nutrient cycling and microbial activity, thereby constraining yield formation [21]. Over the four years, RTSM increased the spike number, grain number per spike, 1000-grain weight, and harvest index, increasing by 14.8%, 10.1%, 7.5%, and 3.6%, respectively, compared to RTNS. These gains may be ascribed to the significant improvements in water and nutrient uptake efficiency induced by rotary tillage [9], and the improved soil moisture retention, reducing evaporation, and stabilizing soil temperature fluctuations by straw mulching [40,41]. The increase in 1000-grain weight and spike number under RTSM was especially notable, as these yield components are highly sensitive to water availability and nutrient supply during the grain-filling stage [42,43]. In contrast, PTSM also increased wheat yield, but the yield increase was lower than that of RTSM relative to their respective non-mulched treatments. The smaller yield improvements observed under PTNS may be due to the limited soil loosening and root penetration compared to RTNS, which restricts water and nutrient uptake efficiency [44,45]. This study highlights the significant enhancement of straw management on wheat yield and its components, with a particularly strong interaction between straw management and tillage methods. The rotary tillage combined with straw mulching gains the highest yield and should be applied in the wheat-soybean double-cropping system.

### 3.2. Effects of Tillage Methods and Straw Management on Dry Matter Accumulation, Translocation, and Distribution

Efficient dry matter accumulation, translocation, and distribution are pivotal for wheat yield in rain-fed cropping systems, driven by tillage method and straw management interactions [5]. Previous studies indicated the significant impact of tillage methods and straw management on dry matter dynamics, and demonstrated that the combination of rotary tillage with straw mulching optimized soil conditions, enhancing nutrient availability, and improving moisture retention, thereby promoting dry matter accumulation and its efficient translocation to grains [5,6,43]. Our results showed that rotary tillage with straw mulching significantly increased dry matter accumulation at anthesis and maturity by 6.1% and 3.7%, respectively, compared to plowing with straw mulching. This result was in accordance with Zhai et al. [46], who reported that rotary tillage with straw mulching enhanced root growth and nutrient uptake, leading to higher dry matter accumulation and grain yield. Furthermore, our trial showed rotary tillage with straw mulching enhanced pre-anthesis dry matter translocation, translocation rate, and contribution to grain yield by 105.0%, 88.4%, and 98.0%, respectively, compared to rotary tillage with no straw mulching. The elevated pre-anthesis dry matter translocation, translocation rate, and the contribution of dry matter translocation to grain under straw mulching may be mainly associated with two synergistic drivers: (1) the continuous release of N-P nutrients during straw decomposition directly enhanced the source–sink capacity during grain filling [47,48]; (2) Rotary tillage reduced soil bulk density in the 12–15 cm layer while increasing root biomass, thereby optimizing root–soil interfaces for assimilate transport [49,50]. These improvements suggest that rotary tillage with straw mulching facilitates a more efficient source–sink relationship, ensuring that assimilates produced before anthesis are effectively re-mobilized to grains. In contrast, under no straw mulching, rotary tillage did not enhance dry matter translocation and even reduced its translocation rate compared to plowing, underscoring the significance of straw returning in optimizing soil properties and crop water–nutrient use efficiency [49]. Yue et al. [51] also reported that straw mulching improves dry matter accumulation and translocation at all growth stages of wheat. In our trials, rotary tillage with straw mulching increased post-anthesis dry matter accumulation by 23.9% compared to no straw mulching, further reinforcing its role in optimizing yield formation.

In addition to above-ground dry matter accumulation, this study also examined dry matter distribution in distinct organs and found that there were significant interactions of tillage methods and straw management. Rotary tillage with straw mulching led to greater dry matter distribution to grains. These results may be explained by the effects of tillage method and straw management on soil physical properties and microbial characteristics [52]. For example, Zhang et al. [53] found that plowing with straw mulching increased soil organic matter content, improving soil nutrient availability, thus encouraging deeper root penetration, dry matter translocation efficiency, supporting grain filling, and finally enhancing dry matter distribution in grains. These findings emphasize the potential of sustainable tillage and straw management in improving dry matter production and ensuring stable wheat yields in rain-fed systems.

### 3.3. Effects of Tillage Methods and Straw Management on Soil Water and Wheat WUE

Proper tillage methods and straw management modify soil physical and chemical properties, thereby increasing soil water storage and crop WUE [5,54]. In this study, although tillage methods and the interaction of tillage methods and straw management had no significant effects on soil water consumption during growth stages, tillage methods and straw management affected the soil water storage in the rain-fed wheat-soybean double-cropping system. Specifically, under RTNS under no straw mulching conditions, rotary tillage increased water storage in the upper soil layers (0–40 cm) at sowing and harvest, while plowing enhanced water storage in deeper soil layers (40–200 cm) at harvest. These findings indicated that, under no straw mulching, rotary tillage maintained higher soil water storage in topsoil, while plowing increased in subsoil. The reason may be due to rotary tillage mainly loosening surface soil porosity (0–20 cm), resulting in water retention mainly existing in the upper soil [55]. When straw mulching is compared to no straw mulching, the soil water storage increased in the 0–40 cm soil layer and decreased in the 40–200 cm soil layer. This enhancement of the topsoil water storage could be attributed to the increased water retention and reduced evaporation [40,56]. Zhao et al. [57] demonstrated that straw mulching effectively improved soil water storage across diverse soil types. A global meta-analysis also showed that, under rotary tillage, straw mulching increased water storage by 5.0% compared to no straw mulching [58]. The decrease in subsoil water storage may be ascribed to the more water uptake by wheat roots induced by straw mulching [59].

The integration of tillage methods and straw management substantially improved WUE [27]. In the present study, PTSM and RTSM increased WUE by 7.4% and 10.4%, respectively, compared to their relative no-straw-mulching treatments. The improved WUE under RTSM was mainly attributed to the enhanced dry matter accumulation (Figure 1) and dry matter translocation to grains (Table 3), ultimately leading to a higher grain yield (Table 1). These results, consistent with a previous study [60], demonstrated that integrating rotary tillage and straw mulching optimizes soil moisture conditions, offering a promising strategy for achieving high-yield wheat production in a rain-fed wheat-soybean double-cropping system. Despite slightly higher water consumption under straw mulching treatments, these increases were offset by the enhanced dry matter production and higher grain yield, and finally enhancing WUE (Figure 3). These findings align with previous research and emphasize that straw mulching is a crucial strategy for improving soil moisture, maximizing WUE, and achieving sustainable yield increases in water-limited environments [18,56,57]. Thus, integrating straw mulching with an appropriate tillage method presents a viable long-term solution for enhancing WUE and stabilizing wheat yield in rain-fed agricultural systems.

Our trial also indicated that the effectiveness of straw mulching on water-saving and yield improvement varied with tillage methods. The straw mulching-induced water-saving amount and rate per kg yield, as well as the yield improvement amount and rate per mm water consumption under rotary tillage, were 223.7%, 195.5%, 187.5%, and 163.3% higher than plowing. This may be due to the pre-sowing soil water storage, yield, and dry matter improvement of straw mulching under rotary tillage being significantly higher than those under plowing in 2, 4, and 2 years, respectively, with a 4-year average increase of 66.0%, 84.5%, and 60.7%. Gao et al. [58] also reported that, in the Loess Plateau, plastic mulching significantly increased the water-saving of rain-fed wheat.

### 3.4. Pathway Analysis Using PLSPM and Comprehensive Evaluation

Regression analysis further underscored that a significant positive correlation was observed between water consumption and dry matter accumulation (*p* < 0.05). The strong association with dry matter accumulation confirms the importance of water availability for crop growth, indicating that adequate moisture is essential for dry matter production [61]. However, the aggregated data did not reveal a statistically significant relationship between water consumption and yield, which is likely due to interannual variability [61]. PLS-PM analysis further demonstrated that straw mulching directly contributed to higher yields through multiple pathways. Straw mulching had a strong positive effect on yield (PC = 0.308, *p* < 0.01), with increased WUE (PC = 0.323, *p* < 0.01) and water consumption (PC = 0.766, *p* < 0.01) significantly improving both dry matter accumulation and grain yield. Furthermore, increased dry matter accumulation directly enhanced yield formation (PC = 0.612, *p* < 0.01), reinforcing the critical role of optimized soil moisture conditions in supporting grain development.

The TOPSIS model revealed that RTSM treatment consistently achieved the highest comprehensive evaluation values over the four-year period, while RTNS recorded the lowest. This outcome is likely due to the superior integration of straw mulching in RTSM, which promoted a more uniform distribution of organic matter, enhancing soil structure, moisture retention, and nutrient recycling [8]. In contrast, RTNS may lead to greater disruption of soil aggregates, resulting in lower organic matter retention and reduced nutrient availability [62]. The intermediate performance of other studies further supports the notion that optimal tillage method and straw management can significantly influence soil properties and crop productivity [48,51]. These mechanistic insights not only validate the TOPSIS results but also emphasize the critical role of straw returning in sustaining long-term agricultural productivity.

## 4. Materials and Methods

### 4.1. Experimental Site Description

This study was conducted from October 2014 to May 2018 based on a long-term located experiment, which was initiated in October 2009, for the wheat-soybean double-cropping system. The experimental site was at the experimental station of Henan University of Science and Technology in Luoyang, Henan Province, China (112.25° E, 34.36° N). The soil at the experimental site is classified as loam with a soil pH of 8.1, organic matter content of 15.9 g kg^−1^, available nitrogen content of 36.3 mg kg^−1^, available phosphorus content of 21.0 mg·kg^−1^, and available potassium content of 120.0 mg kg^−1^ in the 0–20 cm layer at the initiation of the experiment (October 2009). The monthly precipitation and temperature at the experiment site in 2014–2018 are shown in Figure 6. The wheat-growing seasons (from October to May) in 2014–2018 exhibited significant climate variability. Drought years (2014–2015) received 13.2% less precipitation than the 30-year average (244.2 mm). Additionally, extreme drought occurred in November–December 2017 (2 mm/month). While summer rainfall was abundant (e.g., 310 mm in July–August), winter precipitation remained insufficient in 2016–2017, resulting in heightened sensitivity of wheat to water availability during the early growth stage. Notably, during November–February, the 2015–2016 and 2017–2018 seasons experienced temperature reductions of 26.5–82.3% and 12.9–94.1%, respectively, compared to the 30-year average, which may potentially disrupt normal wheat growth.

### 4.2. Experimental Design and Field Management

The experiment was conducted in October 2009 using a split-plot design with the tillage method as the main plot treatment and straw management as the subplot treatment, with three replications each. The two tillage methods were plowing (PT) and rotary tillage (RT). The two straw management methods were no straw mulching (NS) and straw mulching (SM). Thus, four treatments were laid out in the experiment: plowing with no straw mulching (PTNS), plowing with straw mulching (PTSM), rotary tillage with no straw mulching (RTNS), and rotary tillage with straw mulching (RTSM). The specific operations are listed in Table 6.

The plot area was 60 m^2^ (20 m × 3 m), and the management practices in all plots were consistent across all seasons from 2009 to 2018. Wheat cultivar ‘Luohan 6’ and soybean cultivar ‘Zhonghuang 13’ were used. Wheat was sown in middle or late October at a seeding rate of 180.0 kg ha^−1^ and harvested in late May or early June. Soybean was sown in early or middle June at a plant density of 120,000 plants ha^−1^ and harvested in late September or early October. The seeds according to the designed amount were sown using a crop seeder. The row spaces of wheat and soybean were 20 cm and 40 cm, respectively. There was no irrigation during the whole experimental period. Fertilizers (N:P_2_O_5_:K_2_O = 20:15:10) with the amount of 900 kg ha^−1^ for wheat and 300 kg ha^−1^ for soybean were applied as basal. Weeds, pests, and diseases were controlled with herbicides and pesticides according to local practices.

### 4.3. Measurements and Methods

#### 4.3.1. Dry Matter Accumulation, Translocation, and Distribution

At the jointing, anthesis, and maturity stages of wheat, 50 plants were collected from three distinct rows in each plot. After cutting off the root, samples were separated into three components in terms of stem + leaf, glume + rachis, and grain. Samples were immediately oven-dried at 105 °C for 30 min, followed by drying at 65 °C to a constant weight, to determine the dry weight in each organ. The above-ground dry matter accumulation (kg ha^−1^) was the sum of each above-ground organ. The dry matter accumulation, translocation amount, translocation rate, and contribution rate were calculated using the method of Moradi et al. [63]. The dry matter distribution was determined using the method of Cai et al. [64]. The calculations were as follows:

Pre-anthesis dry matter translocation (kg ha^−1^) = dry matter accumulation in above-ground vegetative organs at anthesis − dry matter accumulation in above-ground vegetative organs at maturity.

Pre-anthesis dry matter translocation rate (%) = pre-anthesis dry matter translocation/above-ground dry matter accumulation at anthesis × 100.

Contribution rate of pre-anthesis translocation to grain (%) = pre-anthesis dry matter translocation amount/grain dry matter accumulation at maturity × 100.

Post-anthesis dry matter accumulation (kg ha^−1^) = above-ground dry matter accumulation at maturity − above-ground dry matter accumulation at anthesis.

Contribution rate of post-anthesis dry matter to grain (%) = post-anthesis dry matter accumulation/grain dry matter accumulation at maturity × 100.

Dry matter distribution = DAo/Dat, where DAo is dry matter accumulation in the organ and DAt is dry matter accumulation in the above-ground part.

#### 4.3.2. Grain Yield, Yield Components, and Harvest Index

At the maturity stage in 2014–2018, three 1 m^2^ quadrats were randomly harvested from each plot, then threshed after being air-dried for 3–5 days. Grain samples from the three quadrats were pooled and weighted. A total of 50 ± 5 g air-dried grains were further oven-dried at a temperature of 65 °C for a duration of 24 h to determine the water content in grains. The grain yield for each plot was standardized to a uniform moisture content of 12.5%. Meanwhile, the number of spikes from two random 1 m^2^ areas in each plot was counted to determine spike numbers per hectare, and 30 spikes were sampled to measure the grains per spike and thousand-grain weight. The harvest index was calculated by the grain dry weight relative to above-ground dry matter accumulation at maturity.

#### 4.3.3. Soil Water Storage and WUE

At sowing and maturity stages, two random soil samples were taken to a total depth of 200 cm in 20 cm increments using an auger (inner diameter = 4.0 cm) in each plot. Soil samples from the same layer in each plot were thoroughly merged, and about 300 g of the merged soil samples were carried out for soil analyses. The soil water content was determined gravimetrically by drying in an oven at 105 °C for 24 h. Soil water storage and WUE were calculated using the formulas provided by Li et al. [5]:

Soil water storage (W, mm) = h_i_ × ρ_i_ × ω_i_ × 10, where h is the soil layer depth (cm), ρ is the soil bulk density (g·cm^−3^), ω is the soil water content (%), i represents the i-th soil layer, and 10 is a conversion coefficient.

Water consumption (ET, mm) = P + W1 − W2, where W1 (mm) and W2 (mm) represent 0–100 cm soil water storage before sowing and maturity, respectively, and P (mm) is precipitation during the growth period.

WUE (WUE, kg ha^−1^ mm^−1^) = Y/ET, where Y is the grain yield (kg ha^−1^).

#### 4.3.4. Effectiveness of Straw Mulching on Water-Saving and Yield Productivity

The improvements of straw mulching on the water consumption in the 0–200 cm soil layer, grain yield, and dry matter were the difference between straw mulching and no straw mulching under the same tillage method.

The effectiveness of straw mulching on water-saving was calculated by comparing the water-saving effect per kg of yield, and that on yield improvement was calculated by comparing the yield increase per mm of water consumption during the wheat growth season, according to Gao et al. [58].

Water-saving amount per kg yield (WSA, mm) = ET_NS_/Y_NS_ − ET_SM_/Y_SM_

Water-saving rate per kg yield (WSR %) = WS × Y_NS_/ET_NS_ × 100

Yield-increase amount per mm water consumption (YIA, kg ha^−1^) = Y_SM_/ET_SM_ − Y_NS_/ET_NS_

Yield increase rate per mm water consumption (YIR, %) = ΔY × ET_NS_/Y_NS_) × 100%, where ET_NS_ (mm) and ET_SM_ (mm) are the values of water consumption under no straw mulching and straw mulching, respectively, and Y_NS_ (kg ha^−1^) and Y_SM_ (kg ha^−1^) are the corresponding yield under no straw mulching and straw mulching, respectively.

#### 4.3.5. Calculation of Comprehensive Evaluation Value

Evaluation indicators—such as yield and yield components, dry matter accumulation and translocation, and WUE—vary in units and cannot be directly compared. Therefore, the data for each indicator were first normalized to eliminate dimensional differences. Subsequently, the entropy weight method was applied for objective weighting according to Zou et al. [65]. Finally, the Technique for Order Preference by Similarity to Ideal Solution (TOPSIS) method was used to calculate a comprehensive evaluation value (Ci, 0 < Ci < 1) for each treatment by measuring the distance from the ideal solution; a value closer to 1 indicates that the scheme is more conducive to achieving high wheat yield.

### 4.4. Statistical Analysis

All statistical analyses were performed using SPSS 26.0 software (IBM Corp., Chicago, IL, USA). Data were first examined for normality using the Shapiro–Wilk test and for homogeneity of variances using Levene’s test. When necessary, data were log- or square root-transformed to meet the assumptions of ANOVA. One-way analysis of variance (ANOVA) was then conducted to test the effects of treatments. Means were compared using Duncan’s multiple range test at a significance level of *p* = 0.05, as it offers higher sensitivity in detecting differences in agronomic field experiments with multiple treatment groups. All data are presented as mean ± standard deviation (SD). Graphs were generated using Origin 2021 (Origin Lab Corp., Northampton, MA, USA) and further improved using Adobe Illustrator 2022 (Adobe Inc., Ireland, San Jose, CA, USA) to enhance clarity and visual quality.

## 5. Conclusions

The results of the present study indicated that the effects of tillage methods on dry matter production, yield, and WUE of wheat in a rain-fed wheat-soybean double-cropping system varied depending on straw management. Specifically, PTNS was superior to RTNS, while RTSM outperformed PTSM. Straw mulching was beneficial for increasing soil water storage and promoting water absorption from deep soil by wheat plants, thereby enhancing dry matter accumulation and its translocation and distribution to grains, optimizing yield components, ultimately increasing wheat yield and WUE by 10.5–20.5% and 7.4–10.4%, respectively. RTSM helped to promote the pre-sowing water storage, dry matter accumulation, and translocation, and offered superior water-saving and yield-improvement effects compared to PTSM, making it suitable for widespread high-yield wheat production in a rain-fed wheat-soybean double-cropping system.

## Figures and Tables

**Figure 1 plants-14-02438-f001:**
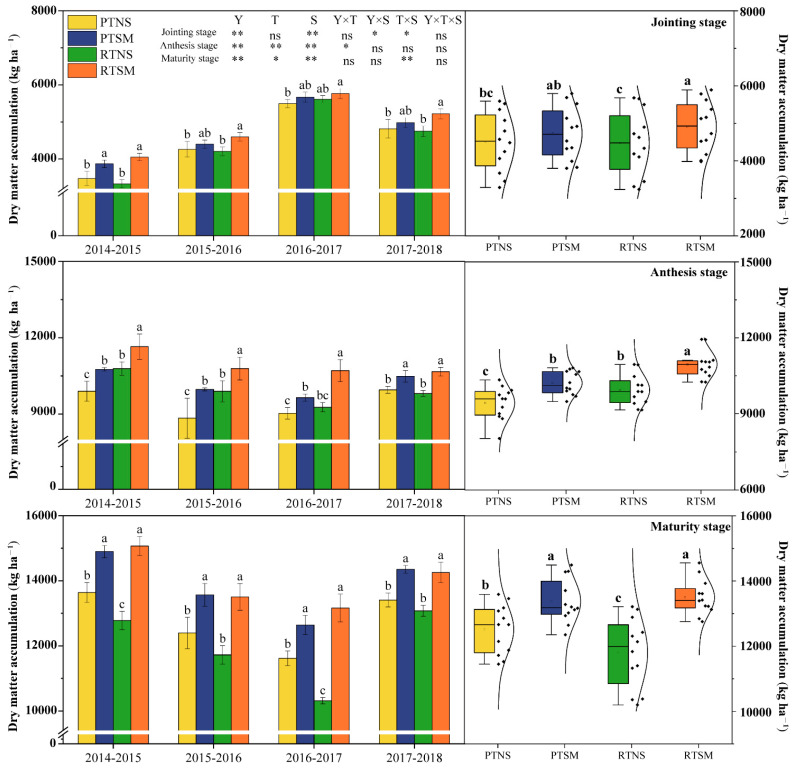
Effects of tillage methods and straw management on dry matter accumulation at jointing, anthesis, and maturity stages of wheat in wheat-soybean double-cropping system. PTNS: plowing with no straw mulching; PTSM: plowing with straw mulching; RTNS: rotary tillage with no straw mulching; RTSM: rotary tillage with straw mulching. Y: years; T: tillage methods; S: straw management. Different lowercase letters within the same year and 4-year average indicate significant difference among treatments (*p* < 0.05). * and ** indicate significant differences at the *p* < 0.05 and *p* < 0.01 levels, respectively. ns means no significance.

**Figure 2 plants-14-02438-f002:**
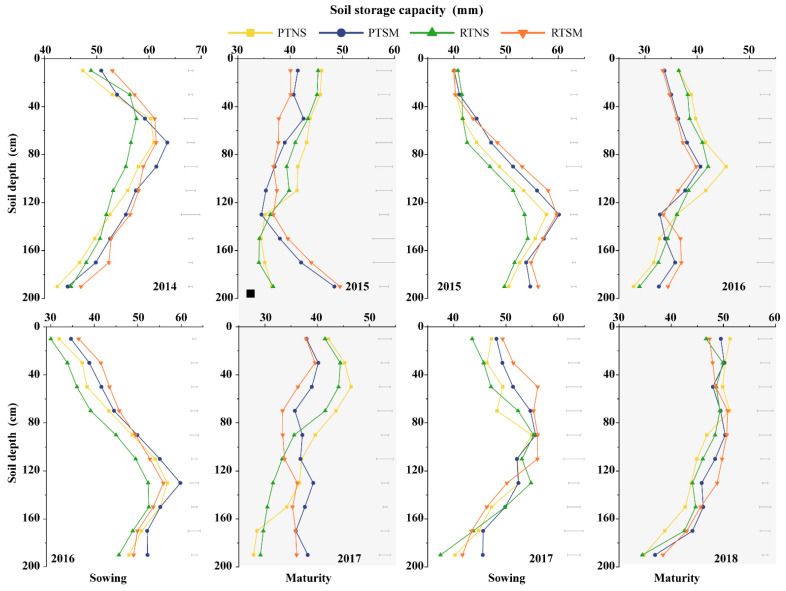
Effects of tillage methods and straw management on soil water storage in 0–200 cm soil layers at sowing and maturity of wheat in wheat-soybean double-cropping system. PTNS: plowing with no straw mulching; PTSM: plowing with straw mulching; RTNS: rotary tillage with no straw mulching; RTSM: rotary tillage with straw mulching. The bars refer to LSD value.

**Figure 3 plants-14-02438-f003:**
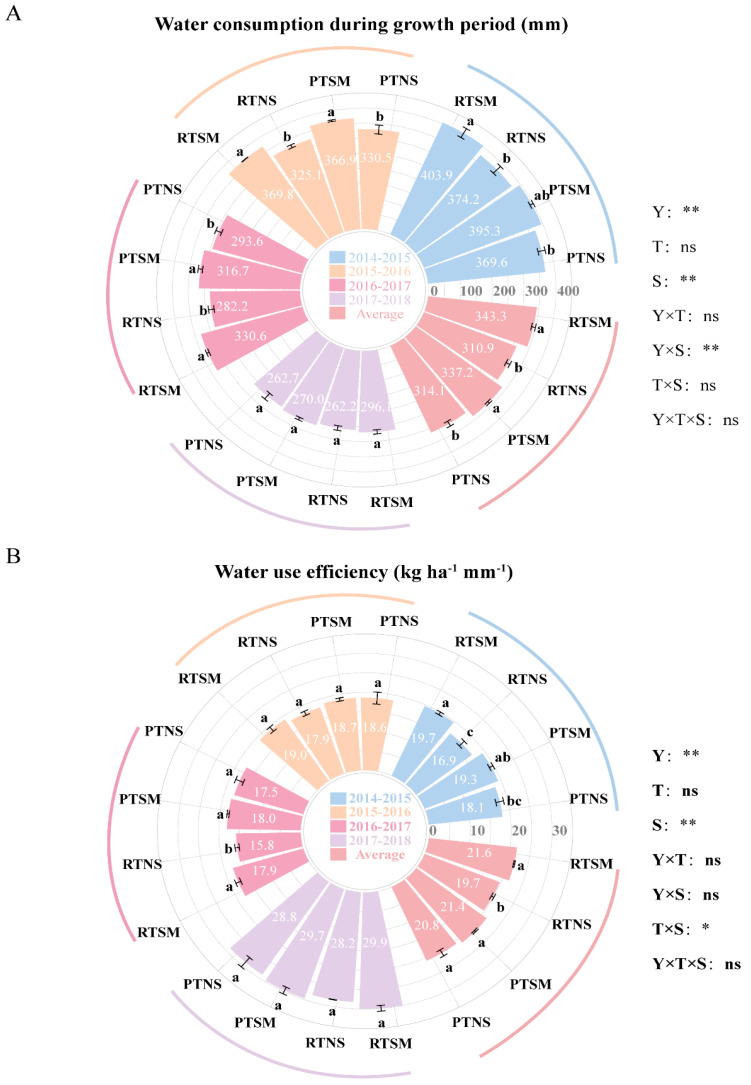
Effects of tillage methods and straw management on water consumption during the wheat growing season (**A**) and WUE (**B**) in wheat-soybean double-cropping system. PTNS: plowing with no straw mulching; PTSM: plowing with straw mulching; RTNS: rotary tillage with no straw mulching; RTSM: rotary tillage with straw mulching. Y: year; T: tillage methods; S: straw management. Different lowercase letters within the same year and 4-year average indicate significant difference among treatments (*p* < 0.05). * and ** indicate significant differences at the *p* < 0.05 and *p* < 0.01 levels, respectively. ns means no significance.

**Figure 4 plants-14-02438-f004:**
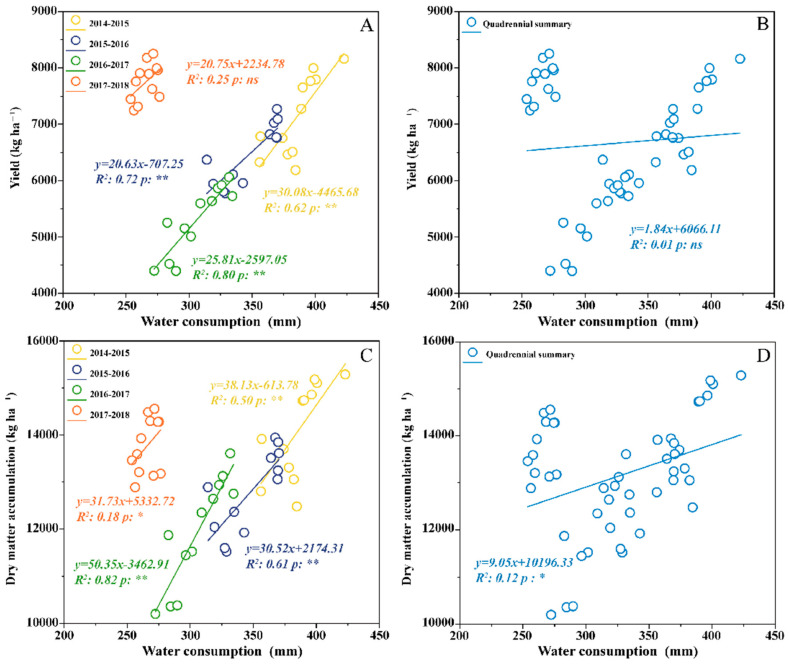
Relationships among grain yield (**A,B**), dry matter accumulation (**C,D**), and water consumption during wheat growing season in wheat-soybean double-cropping system. * and ** indicate significant differences at the *p* < 0.05 and *p* < 0.01 levels, respectively. ns means no significance.

**Figure 5 plants-14-02438-f005:**
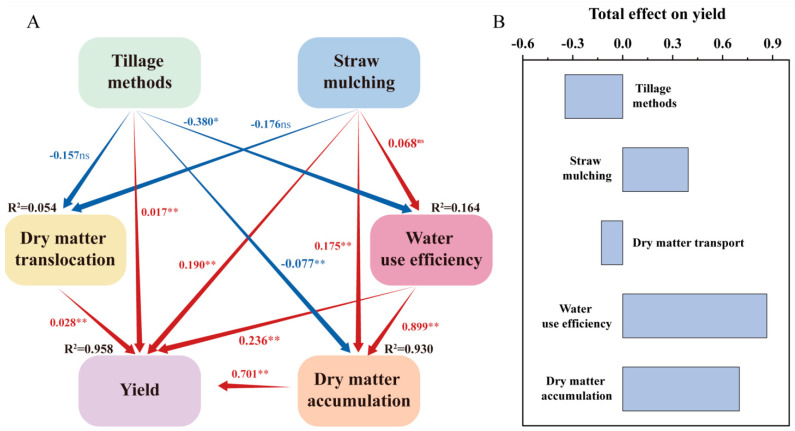
Path diagram of the direct and indirect effects of tillage methods, straw mulching, dry matter accumulation and translocation, and WUE dry matter accumulation on wheat yield (**A**). Total effects of relevant parameters on yield (**B**). * and ** indicate significant differences at the *p* < 0.05 and *p* < 0.01 levels, respectively. ns means no significance.

**Figure 6 plants-14-02438-f006:**
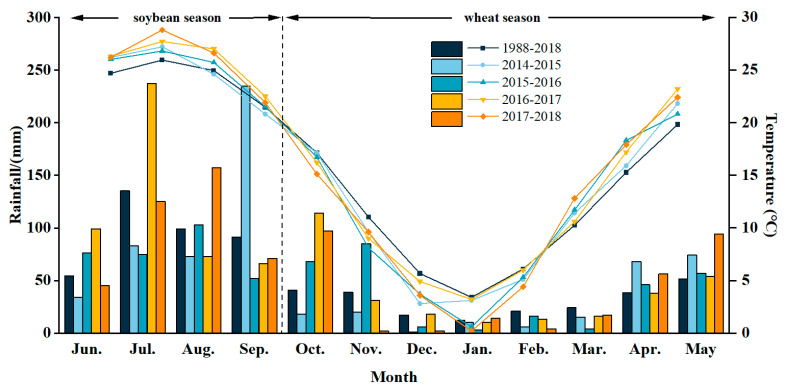
Monthly precipitation and temperature at the experimental site in the experiment years from 2014 to 2018 and the average from 1988 to 2018.

**Table 1 plants-14-02438-t001:** Effects of tillage methods and straw management on grain yield and harvest index of wheat in wheat-soybean double-cropping system.

Years	Treatments	Spike Number (×10^4^ ha^−1^)	Grain Number per Spike	1000-Grain Weight (g)	Grain Yield (kg ha^−1^)	Harvest Index (%)
2014–2015	PTNS	563.3 ± 21.5 ab	33.3 ± 0.1 c	44.5 ± 0.2 c	6668.1 ± 178.4 b	48.9 ± 0.4 c
	PTSM	602.7 ± 34.1 ab	34.6 ± 0.4 b	46.4 ± 0.2 b	7612.9 ± 293.2 a	51.1 ± 1.5 ab
	RTNS	552.0 ± 14.4 b	32.8 ± 0.2 c	44.6 ± 0.1 c	6340.5 ± 161.5 b	49.6 ± 0.3 bc
	RTSM	609.1 ± 16.5 a	35.9 ± 0.7 a	47.0 ± 0.2 a	7937.0 ± 258.7 a	52.7 ± 0.8 a
2015–2016	PTNS	508.0 ± 19.6 b	32.3 ± 0.4 b	43.6 ± 0.2 c	6144.4 ± 209.0 b	49.6 ± 0.3 c
	PTSM	565.7 ± 33.1 a	34.8 ± 0.9 a	45.7 ± 0.4 b	6867.9 ± 140.0 a	50.6 ± 0.3 b
	RTNS	497.1 ± 14.0 b	31.8 ± 0.2 b	42.7 ± 0.1 d	5840.0 ± 90.3 b	49.8 ± 0.4 c
	RTSM	571.9 ± 16.0 a	35.7 ± 0.6 a	46.9 ± 0.2 a	7043.9 ± 254.5 a	52.1 ± 0.4 a
2016–2017	PTNS	427.5 ± 13.9 b	27.4 ± 0.4 b	41.1 ± 0.5 b	5137.9 ± 121.7 b	44.2 ± 0.8 a
	PTSM	484.5 ± 13.9 a	29.3 ± 0.6 a	42.9 ± 0.4 a	5699.2 ± 146.0 a	45.1 ± 0.4 a
	RTNS	419.9 ± 16.5 b	27.1 ± 0.4 b	40.3 ± 1.0 b	4437.9 ± 71.6 c	43.0 ± 0.7 b
	RTSM	508.8 ± 22.1 a	28.9 ± 0.6 a	42.6 ± 0.3 a	5902.5 ± 169.4 a	44.8 ± 0.3 a
2017–2018	PTNS	522.7 ± 19.0 b	32.7 ± 0.3 b	43.2 ± 0.7 b	7564.5 ± 171.0 b	56.4 ± 1.0 a
	PTSM	578.6 ± 32.1 a	34.8 ± 0.6 a	45.1 ± 0.8 a	8011.4 ± 149.6 a	55.8 ± 0.6 a
	RTNS	512.1 ± 13.5 b	32.2 ± 0.2 b	42.3 ± 0.5 b	7396.1 ± 202.3 b	56.6 ± 1.4 a
	RTSM	584.6 ± 15.5 a	35.2 ± 0.6 a	46.2 ± 0.8 a	8049.6 ± 179.7 a	56.5 ± 0.4 a
4-year average	PTNS	505.4 ± 11.7 b	25.1 ± 0.1 b	43.1 ± 0.2 c	6378.7 ± 69.8 b	49.8 ± 0.1 c
	PTSM	557.9 ± 28.2 a	26.7 ± 0.3 a	45.0 ± 0.2 b	7047.8 ± 54.7 a	50.7 ± 0.2 b
	RTNS	495.3 ± 10.0 b	24.7 ± 0.2 b	42.5 ± 0.3 d	6003.6 ± 82.4 c	49.7 ± 0.3 c
	RTSM	568.6 ± 16.0 a	27.2 ± 0.5 a	45.7 ± 0.3 a	7233.2 ± 165.1 a	51.5 ± 0.1 a
ANOVA results	Years (Y)	**	**	*	**	**
	Tillage methods (T)	ns	ns	ns	ns	ns
	Straw management (S)	**	**	**	**	**
	Y × T	ns	ns	ns	ns	*
	Y × S	ns	*	*	**	**
	T × S	ns	**	**	**	*
	Y × T × S	ns	ns	ns	ns	ns

PTNS: plowing with no straw mulching; PTSM: plowing with straw mulching; RTNS: rotary tillage with no straw mulching; RTSM: rotary tillage with straw mulching. Data presented as mean ± SD (*n* = 3). Different small letters within the same year and 4-year average indicate significant differences among treatments (*p* < 0.05). * and ** indicate significant differences at the *p* < 0.05 and *p* < 0.01 levels, respectively. ns means no significance.

**Table 2 plants-14-02438-t002:** Effects of tillage methods and straw management on the characteristics of dry matter accumulation, translocation, and its contribution to grain of wheat in wheat-soybean double-cropping system.

Years	Treatments	Pre-Anthesis Dry Matter			Post-Anthesis Dry Matter	
		Translocation Amount (kg ha^−1^)	Translocation Rate (%)	Contribution Rate (%)	Accumulation (kg ha^−1^)	Contribution Rate (%)
2014–2015	PTNS	2922.4 ± 190.7 c	29.5 ± 0.8 c	43.8 ± 2.2 c	3745.7 ± 140.ab	56.2 ± 2.2 a
	PTSM	3467.5 ± 114.7 b	32.3 ± 1.3 b	45.6 ± 3.3 c	4145.4 ± 401.6 a	54.4 ± 3.3 a
	RTNS	4347.2 ± 173.3 a	40.3 ± 1.5 a	68.6 ± 3.5 a	1993.3 ± 254.7 c	31.4 ± 3.5 c
	RTSM	4514.3 ± 410.5 a	38.7 ± 2.0 a	56.8 ± 3.8 b	3422.7 ± 238.2 b	43.2 ± 3.8 b
2015–2016	PTNS	2587.6 ± 505.1 c	29.1 ± 3.2 b	42.0 ± 6.8 b	3556.8 ± 306.0 a	58.0 ± 6.8 a
	PTSM	3267.9 ± 158.8 b	32.8 ± 1.8 b	47.6 ± 3.3 b	3600.0 ± 297.7 a	52.4 ± 3.3 a
	RTNS	4010.5 ± 297.9 a	40.5 ± 1.7 a	68.7 ± 5.0 a	1829.5 ± 290.6 c	31.3 ± 5.0 b
	RTSM	4326.0 ± 297.8 a	40.1 ± 1.4 a	61.4 ± 3.0 a	2717.9 ± 203.4 b	38.6 ± 3.0 b
2016–2017	PTNS	2547.6 ± 51.2 b	28.2 ± 0.3 c	49.6 ± 0.6 c	2590.3 ± 80.9 b	50.4 ± 0.6 a
	PTSM	2701.4 ± 26.0 b	28.0 ± 0.7 c	47.4 ± 1.5 c	2997.8 ± 162.9 a	52.6 ± 1.5 a
	RTNS	3392.9 ± 142.6 a	36.6 ± 1.6 a	76.5 ± 3.4 a	1045.0 ± 154.7 c	23.5 ± 3.4 c
	RTSM	3446.4 ± 191.9 a	32.2 ± 0.8 b	58.5 ± 4.7 b	2456.1 ± 347.3 b	41.5 ± 4.7 b
2017–2018	PTNS	3266.0 ± 155.3 b	32.9 ± 2.0 a	48.6 ± 3.1 a	3458.0 ± 278.5 ab	51.4 ± 3.1 a
	PTSM	3249.1 ± 75.8 b	31.0 ± 0.8 a	45.6 ± 1.5 a	3872.1 ± 167.2 a	54.4 ± 1.5 a
	RTNS	3306.3 ± 116.9 b	33.7 ± 1.4 a	50.4 ± 3.1 a	3268.1 ± 296.0 b	49.6 ± 3.1 a
	RTSM	3564.7 ± 66.5 a	33.4 ± 1.0 a	49.8 ± 1.6 a	3590.4 ± 182.6 ab	50.2 ± 1.6 a
4-year average	PTNS	2830.9 ± 150.5 c	30.0 ± 0.4 c	46.0 ± 1.3 c	3337.7 ± 28.3 b	54.1 ± 1.3 a
	PTSM	3171.5 ± 74.4 b	31.1 ± 1.1 c	46.6 ± 2.1 c	3653.8 ± 237.1 a	53.5 ± 2.2 a
	RTNS	3764.2 ± 86.2 a	37.9 ± 1.0 a	66.0 ± 2.1 a	2034.0 ± 171.2 c	35.1 ± 2.3 c
	RTSM	3962.9 ± 206.9 a	36.1 ± 0.9 b	56.6 ± 2.2 b	3046.8 ± 125.0 b	43.5 ± 2.2 b
ANOVA results	Years (Y)	**	**	**	**	**
	Tillage methods (T)	**	**	**	**	**
	Straw management (S)	**	ns	**	**	**
	Y × T	**	**	**	**	**
	Y × S	ns	*	*	*	*
	T × S	ns	**	**	**	**
	Y × T × S	ns	ns	*	*	*

PTNS: plowing with no straw mulching; PTSM: plowing with straw mulching; RTNS: rotary tillage with no straw mulching; RTSM: rotary tillage with straw mulching. Data presented as mean ± SD (*n* = 3). Different small letters within the same year and 4-year average indicate significant differences among treatments (*p* < 0.05). * and ** indicate significant differences at the *p* < 0.05 and *p* < 0.01 levels, respectively. ns means no significance.

**Table 3 plants-14-02438-t003:** Effects of tillage methods and straw management on dry matter distribution at maturity of wheat in wheat-soybean double-cropping system.

Years	Tillage Methods	Stem + Leaf		Glume		Grain	
		DMD (kg ha^−1^)	Percentage (%)	DMD (kg ha^−1^)	Percentage (%)	DMD (kg ha^−1^)	Percentage (%)
2014	PTNS	5097.7 ± 140.4 a	37.4 ± 0.3 a	1873.7 ± 64.7 ab	13.7 ± 0.4 a	6668.1 ± 178.4 b	48.9 ± 0.4 c
	PTSM	5298.7 ± 132.1 a	35.6 ± 1.1 ab	1986.7 ± 52.0 a	13.3 ± 0.4 a	7612.9 ± 293.2 a	51.1 ± 1.5 ab
	RTNS	4695.3 ± 132.2 b	36.8 ± 0.5 a	1741.6 ± 120.8 b	13.6 ± 0.8 a	6340.5 ± 161.5 b	49.6 ± 0.3 bc
	RTSM	5191.7 ± 89.4 a	34.5 ± 0.7 b	1941.4 ± 56.6 a	12.9 ± 0.3 a	7937.0 ± 258.7 a	52.7 ± 0.8 a
2015	PTNS	4600.7 ± 216.0 a	37.1 ± 0.2 a	1649.3 ± 59.0 bc	13.3 ± 0.1 a	6144.4 ± 209.0 b	49.6 ± 0.3 c
	PTSM	4852.0 ± 122.4 a	35.8 ± 0.1 b	1846.4 ± 98.7 a	13.6 ± 0.4 a	6867.9 ± 140.0 a	50.6 ± 0.3 b
	RTNS	4293.7 ± 122.6 b	36.6 ± 0.3 a	1588.6 ± 78.2 c	13.6 ± 0.4 a	5840.0 ± 90.3 b	49.8 ± 0.4 c
	RTSM	4688.0 ± 160.0 a	34.7 ± 0.3 c	1773.6 ± 52.4 ab	13.2 ± 0.6 a	7043.9 ± 254.5 a	52.1 ± 0.4 a
2016	PTNS	4933.7 ± 92.2 b	42.5 ± 0.3 a	1544.1 ± 96.2 b	13.3 ± 0.6 a	5137.9 ± 121.7 b	44.2 ± 0.8 a
	PTSM	5312.3 ± 144.3 a	42 ± 0.5 a	1630.5 ± 30.2 ab	12.9 ± 0.1 a	5699.2 ± 146.0 a	45.1 ± 0.4 a
	RTNS	4416.0 ± 103.7 c	42.8 ± 0.3 a	1457.3 ± 119.4 b	14.1 ± 1.2 a	4437.9 ± 71.6 c	43 ± 0.7 b
	RTSM	5462.0 ± 91.8 a	41.5 ± 0.7 a	1797.8 ± 172.6 a	13.7 ± 0.8 a	5902.5 ± 169.4 a	44.8 ± 0.3 a
2017	PTNS	4802.1 ± 213.6 bc	35.8 ± 0.6 a	1882.6 ± 69.7 b	14.1 ± 0.8 a	6724.0 ± 152.0 b	50.2 ± 0.9 a
	PTSM	5140.7 ± 127.8 a	35.8 ± 0.8 a	2093.3 ± 87.2 a	14.6 ± 0.6 a	7121.2 ± 133.0 a	49.6 ± 0.6 a
	RTNS	4610.2 ± 158.8 c	35.2 ± 1.1 a	1891.7 ± 47.3 b	14.5 ± 0.2 a	6574.3 ± 179.8 b	50.3 ± 1.2 a
	RTSM	5034.8 ± 163.0 ab	35.3 ± 0.6 a	2064.1 ± 49.8 a	14.5 ± 0.3 a	7155.2 ± 159.7 a	50.2 ± 0.3 a
4-year	PTNS	4858.5 ± 115.8 b	38.0 ± 0.2 a	1737.4 ± 11.3 b	13.6 ± 0.3 a	6168.6 ± 69.5 b	48.3 ± 0.1 c
	PTSM	5150.9 ± 13.4 a	37.2 ± 0.1 ab	1889.2 ± 51.7 a	13.6 ± 0.3 a	6825.3 ± 55.9 a	49.2 ± 0.3 b
	RTNS	4503.8 ± 78.7 c	37.6 ± 0.3 a	1669.8 ± 47.2 b	13.9 ± 0.4 a	5798.2 ± 78.3 c	48.4 ± 0.3 c
	RTSM	5094.1 ± 87.7 a	36.4 ± 0.5 b	1894.2 ± 82.8 a	13.5 ± 0.5 a	7009.6 ± 160.4 a	50.1 ± 0.2 a
ANOVA results	Years (Y)	**	**	**	**	**	**
	Tillage modes (T)	**	*	ns	ns	ns	*
	Straw management (S)	**	**	**	ns	**	**
	Y × T	ns	ns	ns	ns	ns	**
	Y × S	**	*	ns	ns	**	**
	T × S	**	ns	ns	ns	**	*
	Y × T × S	ns	ns	ns	ns	ns	ns

PTNS: plowing with no straw mulching; PTSM: plowing with straw mulching; RTNS: rotary tillage with no straw mulching; RTSM: rotary tillage with straw mulching. DMD: dry matter distribution. Data presented as mean ± SD (*n* = 3). Different small letters within the same year and 4-year average indicate significant difference among treatments (*p* < 0.05). * and ** indicate significant differences at the *p* < 0.05 and *p* < 0.01 levels, respectively. ns means no significance.

**Table 4 plants-14-02438-t004:** Effects of straw mulching on water-saving and productivity improvement of wheat in wheat-soybean double-cropping system.

Years	Tillage Methods	Water Storage Improvement (mm)	Yield Improvement (kg ha^−1^)	Dry Matter Improvement (kg ha^−1^)	Water-Saving per kg Yield		Yield Improvement per mm Water Consumption	
					Amount (mm)	Rate (%)	Amount (kg ha^−1^)	Rate (%)
2014–2015	PT	22.0 ± 0.6 a	944.7 ± 115.3 b	2417.4 ± 679.5 a	3.5 ± 1.5 b	6.3 ± 2.8 b	1.2 ± 0.6 b	21.8 ± 9.9 b
	RT	34.5 ± 2.3 a	1596.6 ± 112.3 a	3348.2 ± 837.7 a	8.1 ± 2.1 a	13.7 ± 3.3 a	2.7 ± 0.7 a	45.7 ± 10.6 a
2015–2016	PT	20.9 ± 2.4 b	723.5 ± 73.1 b	2426.6 ± 1186.5 a	1.1 ± 0.1 b	2.1 ± 0.3 b	0.4 ± 0.1 b	7.2 ± 1.0 b
	RT	37.4 ± 1.1 a	1203.9 ± 182.2 a	3052.3 ± 742.3 a	3.1 ± 2.1 b	5.6 ± 3.7 b	1.1 ± 0.7 b	19.4 ± 13.1 b
2016–2017	PT	19.7 ± 2.1 b	561.3 ± 67.9 b	1818.6 ± 217.8 b	1.6 ± 0.7 b	2.7 ± 1.2 b	0.5 ± 0.2 b	8.6 ± 3.9 b
	RT	44.9 ± 5.5 a	1464.6 ± 116.1 a	4446.7 ± 551.5 a	7.5 ± 2.2 a	11.8 ± 3.2 a	2.1 ± 0.6 a	33.2 ± 8.7 a
2017–2018	PT	21.6 ± 1.0 a	397.2 ± 24.6 b	1644.9 ± 416.9 b	1.1 ± 1 b	2.9 ± 2.4 b	0.8 ± 0.6 b	19.3 ± 15.8 b
	RT	22.9 ± 5.3 a	580.8 ± 24.1 a	2504.8 ± 205.4 a	2.3 ± 0.5 b	5.7 ± 1.3 b	1.5 ± 0.4 b	38.2 ± 9.4 b
4-year average	PT	21.0 ± 1.0 b	656.7 ± 7.1 b	2076.9 ± 488.2 b	1.6 ± 1 b	3.1 ± 2.0 b	0.6 ± 0.4 b	13.0 ± 8.7 b
	RT	34.9 ± 2.7 a	1211.5 ± 101.3 a	3338 ± 572.8 a	5.3 ± 1.3 a	9.2 ± 2.0 a	1.9 ± 0.4 a	34.1 ± 5.5 a
ANOVA results	Years (Y)	**	**	ns	**	**	**	*
	Tillage method (T)	**	**	**	**	**	**	**
	Y × T	**	**	ns	*	ns	ns	ns

PT: plowing; RT: rotary tillage. Data presented as mean ± SD (*n* = 3). Values followed by different lowercase letters within the same year and 4-year average indicate significant difference among treatments (*p* < 0.05). * and ** indicate significant differences at the *p* < 0.05 and *p* < 0.01 levels, respectively. ns means no significance.

**Table 5 plants-14-02438-t005:** The degree of fit and ranking under different treatments by TOPSIS method.

Treatments	2014–2015				2015–2016				2016–2017				2017–2018			
	di+	di−	di	Ranking	di+	di−	di	Ranking	di+	di−	di	Ranking	di+	di−	di	Ranking
PTNS	0.24	0.14	0.38 c	3	0.26	0.16	0.38 c	3	0.20	0.18	0.47 c	3	0.23	0.16	0.41 c	3
PTSM	0.11	0.25	0.69 b	2	0.17	0.22	0.57 b	2	0.12	0.25	0.68 b	2	0.15	0.23	0.60 b	2
RTNS	0.29	0.09	0.24 d	4	0.29	0.10	0.26 d	4	0.28	0.11	0.29 d	4	0.26	0.11	0.29 d	4
RTSM	0.07	0.30	0.81 a	1	0.12	0.26	0.68 a	1	0.07	0.28	0.80 a	1	0.11	0.26	0.71 a	1

di+: The distance of each evaluation scheme to the positive ideal solution; di−: the distance of each evaluation scheme to the negative ideal solution; Ci: closeness coefficient. Different lowercase letters following the data in the same column indicate significant differences among treatments at the *p* < 0.05 level.

**Table 6 plants-14-02438-t006:** Experimental treatments and operation methods.

**Code**	**Treatment**	**Specific Operation**
PTNS	Plowing with no straw mulching	The straw of the previous crop was removed from the plot 1–3 days before tillage. After evenly broadcasting fertilizers by hand, the plowing (30–35 cm) was carried out using a moldboard plow, and the rotary tillage (12–15 cm) was carried out to smooth land using a rotavator. The same management was applied during both wheat and soybean seasons.
PTSM	Plowing with straw mulching	The field management was the same as PTNS except for the straw stubble (<5 cm) of the previous crop being evenly mulched to the surface of the original plot before emergence of the in-season crop.
RTNS	Rotary tillage with no straw mulching	The straw of the previous crop was removed from the plot 1–3 days before tillage. After evenly broadcasting fertilizers by hand, the rotary tillage (12–15 cm) was carried out using a rotavator. The same management practices were applied during both wheat and soybean seasons.
RTSM	Rotary tillage with straw mulching	The field management was the same as RTNS except for the straw stubble (<5 cm) of the previous crop being evenly mulched to the surface of the original plot before emergence of the in-season crop.

## Data Availability

This study includes all supporting data, which can be obtained from the corresponding authors upon request.

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
