# Peer review of "Long-Term Rotary Tillage and Straw Mulching Enhance Dry Matter Production, Yield, and Water Use Efficiency of Wheat in a Rain-Fed Wheat-Soybean Double Cropping System"

_plants, 2025, doi:10.3390/plants14152438_

Round 1

Reviewer 1 Report

Comments and Suggestions for Authors

The manuscript by Dong et al. intended to identify suitable tillage and straw management practices to improve dry matter production, grain yield and water use efficiency of wheat in the dryland winter wheat-summer bean double cropping system. The research in this aspect help to improve the wheat yield, but the current manuscript still has some drawbacks need to be addressed.

1, The abstract section should be concise, highlighting only the key results and conclusions. Briefly clarify the mechanism by which RTSM enhances the crop yield and water use efficiency.

2, Line 55-69, in this paragraph, the authors need to briefly introduce the differences between rotary tillage and plowing in their impacts on soil physical and chemical properties, so as to enhance the intuitiveness of the comparison between them.

3, Line 96-97, a brief introduction is needed regarding the differences between monocropping systems and double-cropping systems in terms of their impacts on soil and crops.

4, In the Methods section, the reason for conducting the experiment continuously for four years has not been explained. Since the charts and tables list and compare data such as yield and water use across different years, why is there no corresponding explanation, illustration, or discussion provided?

5, Line 341-343, the sentence need to be corrected, there is a grammatical error.

6, The Discussion section is excessively lengthy, it does not need to repeat the comparison and analysis of results; instead, it should be condensed and focused on discussions to demonstrate the key conclusions.

7, Line 457-459, since the enhancing soil penetration resistance will hinder root penetration and reduce the mobility of water and nutrients in the soil, this will increase the difficulty for roots to absorb water and nutrients. There is a contradiction in the expression of this sentence, the authors need to revise it to ensure accurate and reasonable explanation.

8, Line 525-527, the statement that "increased soil water storage" and "facilitate water infiltration" are mutually contradictory. At least, the authors should explain in more detail, for instance, in which soil layer the water storage was increased, and in which layer the water infiltration was enhanced.

Reviewer 2 Report

Comments and Suggestions for Authors This paper is in line with the research direction of the discipline, with good innovation in content, sufficient workload, and certain agronomic value in the results. There are the following deficiencies, and revisions are suggested:

  1. Table 1 mentions the maize growing season, but there is no mention of maize in the text. Could this be an error?
  2. How is the straw returned? The respective treatments of soybean straw and wheat straw are not mentioned in the text.
  3. In Section 3.4, the authors are requested to clarify whether it is discussing the effect of straw returning or the effect of tillage. Including the figure caption, there seems to be an error in this part.

Reviewer 3 Report

Comments and Suggestions for Authors

This manuscript presents results from a multi-year field experiment evaluating the effects of rotary tillage and straw mulching on dry matter production, yield, and water use efficiency (WUE) of wheat in a rain-fed wheat–soybean double cropping system. The topic is relevant for sustainable crop management in dryland agriculture, and the long-term dataset provides valuable insights. The experimental design is sound, and the data are rich and comprehensive. However, the manuscript requires important revision before it can be considered for publication. Several sections suffer from repetition, speculative interpretation, and structural imbalances. Especially the discussion part, tends to overextend into mechanistic inferences that are not supported by the data from study. In addition, figures and tables, while extensive, are often poorly formatted or lack sufficient explanation in the main text. Finally, the quality of English requires also moderate to major revision to meet the standards of the journal. Below are my comments.

  1. Lines 13–15: The abstract states the experiment began in 2009, yet results are reported from 2014 to 2018. The rationale for the five-year gap between initiation and analysis must be clarified. Was there an establishment phase, or were previous years excluded for methodological reasons?
  2. Lines 55–69: The discussion of plowing vs. rotary tillage in the introduction is somewhat contradictory. Some studies are cited in support of plowing, others in favor of rotary tillage, without clearly framing the unresolved knowledge gap. This section would benefit from a more critical synthesis to justify the present study’s novelty and the hypothesis being tested.
  3. Lines 122–124: Again, here the authors state the study was conducted at a site established in 2009, but the main experimental details (design, treatments, replications) appear to start from 2014. The long-term context should be better integrated e.g., whether previous years influenced soil conditions, and whether rotations and residue carryover were monitored across seasons.
  4. Table 1: The table lacks clarity in distinguishing between the treatments. The terminology used in the “Specific operation” column is long and repetitive. A cleaner format summarizing the treatment combinations would help readers grasp the four experimental groups more quickly.
  5. Lines 230–237: The statistical analysis section is too brief for a multi-year field experiment with complex interactions. There is no mention of whether the assumptions of ANOVA (normality, homoscedasticity) were verified. The rationale for choosing Duncan’s test should be justified given current recommendations for stronger post-hoc methods.
  6. Figure 4: Figure 4B, the subplots A and B are not described adequately in the figure caption or referred to consistently in the results section. Ensure that all figure panels are described and labeled properly in the text.
  7. Lines 388–405: The use of PLSPM (Partial Least Squares Path Modeling) is not well explained in terms of model validation, fit indices, or variable collinearity. Since the manuscript draws major conclusions from this analysis, the statistical strength of the model must be clearly demonstrated and assumptions stated.
  8. Lines 406–411: The use of TOPSIS is an unusual choice for agronomic field data. The justification for this method must be strengthened. It should also be clarified how weights were assigned in the entropy method and whether the normalization of indices affected rankings. Without clear methodological grounding, these rankings may appear arbitrary.
  9. Discussion Sections 4.1 and 4.2: The discussion relies extensively on biochemical and physiological processes (e.g., lignocellulose hydrolase activity, ethylene-auxin signaling) without presenting experimental evidence from this study. These points should be revised to focus on observed agronomic and soil-related outcomes, or clearly framed as hypothetical with appropriate caution.
  10. Lines 521–536: The role of soil depth in water storage is discussed, but no data are provided on root profiles or deep-water extraction. The claim that rotary tillage improves deep moisture retention is not convincingly supported and should be moderated unless validated with root or soil penetration data.
  11. Throughout Results and Discussion: The treatment effects are repeatedly presented with percent increases (e.g., “RTSM increased yield by 20.5%”), but the magnitude of baseline values is often unclear. Where possible, include absolute values alongside percent changes for better interpretation.
Comments on the Quality of English Language

The manuscript requires moderate to substantial language editing. Grammatical errors, awkward sentence constructions, and non-standard phrasing are frequent throughout the text. Examples include misuse of prepositions (e.g., "increased... for yield"), redundancy ("significantly increased... significantly"), and non-native structures ("yield yield increase"). A professional English language edit is strongly recommended to improve clarity and flow. Additionally, consistent terminology (e.g., "dry matter accumulation" vs. "dry matter production") should be maintained across sections.

Round 2

Reviewer 1 Report

Comments and Suggestions for Authors

Thanks for the authors' hard work on the improvement of this manuscript, they have addressed all my concerns. I have no other question.

Reviewer 3 Report

Comments and Suggestions for Authors

I have no further comments on the manuscript.
Thank you